# Heavy Metals Removal from Domestic Sewage in Batch Mesocosm Constructed Wetlands using Tropical Wetland Plants

Shin Ying Ang [1], Hui Weng Goh [2,*], Bashirah Mohd Fazli [1], Hazzeman Haris [3], Nor Ariza Azizan [2], Nor Azazi Zakaria [2] and Zubaidi Johar [1]

1 National Water Research Institute of Malaysia (NAHRIM), Lot 5377, Jalan Putra Permai, Seri Kembangan 43300, Selangor, Malaysia
2 River Engineering and Urban Drainage Research Centre (REDAC), Universiti Sains Malaysia (USM), Nibong Tebal 14300, Penang, Malaysia
3 School of Biological Sciences, Universiti Sains Malaysia (USM), Penang 11800, Penang, Malaysia
* Correspondence: redac_gohhuiweng@usm.my

**Abstract:** Constructed wetlands are an affordable and reliable green alternative to conventional mechanical systems for treating domestic sewage. This study investigates the potential of 14 tropical wetland plant species for removing heavy metals from domestic sewage through the bioconcentration factor (BCF), translocation factor (TF), enrichment factor (EF), and geoaccumulation index (Igeo) using batch mesocosm studies. Plants with BCF > 1 and TF > 1 are classified as phytoextractors, while species with BCF > 1 and TF < 1 are phytostabilisers. The results indicate that 11 out of 14 species are magnesium phytostabilisers, 10 are calcium phytoextractors, and no plant species demonstrate ferrum phytoextraction properties. As for manganese phytoremediation, only three species depicted phytoextraction and phytostabilisation properties. The enrichment factor (EF) for all of the studied metals with ferum as a reference metal in all of the soil samples decreased after the phytoremediation of domestic sewage experiments, indicating depletion to mineral enrichment (EF < 2). All of the soil samples are generally classified as uncontaminated based on $I_{geo}$ indices. Based on the factors and indices, it is suggested that the plants may have facilitated heavy metal removal from domestic sewage through uptake into the plant tissues from the roots.

**Keywords:** mesocosm study; tropical wetland plants; domestic sewage; heavy metals removal; phytoremediation; constructed wetland; bioconcentration and translocation factor

## 1. Introduction

Domestic sewage or domestic wastewater refers to black and grey water discharged daily from the lavatories, showers, laundries, kitchens, and wash basins of households, restaurants, public facilities, commercials, and other facilities [1–5]. Domestic sewage is characterised by produced volume, flow rate, physical condition, and chemical and toxic constituents. Sewage consists of nutrients (nitrogen and phosphorus), organic matter, inorganic salts, heavy metals, bacteria, and viruses [6]. Domestic sewage contamination poses risks and impacts on human health and natural ecosystems [5,7]. The elevated levels of nutrients in sewage will not only lead to eutrophication in rivers, lakes, and seas [8], but transitional basins and wetlands exposed to domestic sewage will also experience changes in their carbon balance as it can affect the aquatic food web [9] and wetlands' productivity [10]. Sewage is also one of the significant threats to coral reef deterioration worldwide, where nutrients, suspended solids, sediments, pathogens, endocrine disrupters, and heavy metals can impede coral growth and reproduction [7,11]. In Malaysia, sewage pollution has led to ecosystem imbalance in several islands due to elevated nutrient inputs [12]. It has been reported that poorly treated sewage and untreated sullage are directly discharged into drainage system or streams, especially in tourist islands [13]. Apart from this, sewage is one

of Malaysia's most significant pollutant load contributors [14]. Therefore, effectively treating domestic sewage is crucial to solve sewage pollution and to preserve the environment more sustainably, especially in tourist islands and rural areas in Malaysia.

Apart from nutrients, metals or heavy metals are among the known environmental pollutants. Almost all anthropogenic activities contribute to heavy metal accumulation in water bodies [15]. Heavy metals in sewage come from various sources, such as detergents, food, cosmetics, tap water, sweat and dust, and are discharged as wastewater from the lavatory, kitchen, laundry, and bath [14]. Previous studies have revealed that domestic wastewater contains heavy metals, including Fe, Zn, Mn, Cu, Pb, Ni, Cr, Cd, As, and Hg [16–18]. Heavy metal toxicity has proven to be a significant risk to human health [19,20] and can be detrimental even in small quantities [21,22]. Heavy metals cause impairment in neurological, renal, haematological, cardiovascular, and reproductive systems, as well as having adverse effects on the brain, lungs, kidneys, heart, liver, and gastrointestinal tract, and causing respiratory disorders, cancers, and other diseases or disorders in vital organs and cells [19,20,23,24]. However, most metals or heavy metals are essential micronutrients for plants, animals, or humans and ecosystem equilibriums when found in trace amounts [25]. Determining the heavy metals in domestic sewage is substantial for monitoring environmental pollution. Heavy metals can be removed from the surrounding environment by wetland plant species, which can be applied to purify contaminated waters [26].

A constructed wetland is an alternative to conventional sewage treatment plants (STP) that is an affordable and reliable green approach [27,28]. Conventional systems for wastewater treatment require intensive energy for mechanical components with high operational and investment costs. In most developing countries, the current systems for wastewater treatment are failing to treat wastewater adequately because of high costs in terms of operation and maintenance [29]. A constructed wetland is an ecological-based wastewater treatment system that is low in capital, operation, and maintenance [30], and involves plants, soils, and associated microbial assemblages [31]. In constructed wetlands, vegetation is essential for removing nutrients from wastewater and substrates by providing large biofilm surfaces, thus improving the ability to purify water in the rhizosphere [32]. Aquatic plants can uptake excess pollutants such as organic and inorganic, heavy metals, and pharmaceutical pollutants in agricultural, domestic, and industrial wastewater [33].

Many studies have been reported internationally regarding the treatment of various wastewater using constructed wetlands, such as sewage in Thailand [34], sewage in China [35], domestic wastewater in Spain [36] and India [27], and municipal wastewater in China [37]. Constructed wetlands were applied in a Malaysian resort to treat black and grey water [38]. Studies on aquatic plants' performance in the phytoremediation of different wastewater were reviewed by [33], who also reported the application of *Salvinia molesta* plants for the phytoremediation of secondary and tertiary treatment of domestic wastewater [39,40]. The phytoremediation of domestic wastewater in constructed wetlands planted with *Canna x generalis* reeds was reported by [41]. In contrast, the performance of *Canna indica* regarding the treatment of hostel greywater in a constructed wetland in India was studied by [42]. However, there are a lack of comprehensive studies on the phytoremediation properties of domestic sewage using tropical wetland plants in tropical countries.

Many plant species absorb pollutants from soils, such as lead, cadmium, chromium, arsenic, and radionuclides. Phytoremediation through phytoextraction removes metals from the soil by absorbing important metals for plant growth, including Fe, Mn, Zn, Cu, Mg, Mo, and Ni [43]. Examples of vegetation used for heavy metal removal are *Phragmites australis* in constructed wetlands in Italy and France [44,45]; wild plants in Pakistan for Cd, Pb, Cr, Ni, and Cu removal; *Typha latifolia*, *Cyperus alternifolius,* and *Cynodon dactylon* in Nigeria for Cd and Pb removal [46]; and *Phragmites australis* and *Typha latifolia* in Taiwan for Cr, Zn, and Ni removal [47]. However, due to inadequate research on phytoremediation plant studies in the tropics, there is a need to study phytoremediation properties by using tropical plants to remove heavy metals from domestic sewage in columnar mesocosm constructed wetlands, particularly in tropical countries.

This study aims to identify the potential tropical wetland plant species and phytoremediation properties for the removal of heavy metals in domestic sewage using 14 easily available plant species in the tropical climate through the determination of the bioconcentration factor (BCF), translocation factor (TF), soil enrichment factor (EF), and geoaccumulation index ($I_{geo}$) under batch mesocosm column study. The heavy metals were chosen based on their abundance in the domestic sewage discharged from university student hostels. In this study, only Ca, Fe, Mg, and Mn were detected during the initial screening of 21 heavy metal parameters.

## 2. Materials and Methods

### 2.1. Experimental Design

Batch mesocosm experiments were conducted from December 2020 to March 2021 using 45 units of 240 L modified blue storage drums with a diameter of 600 mm and height of 880 mm. The media used in the mesocosms consisted of 100 mm thick gravels, followed by a layer of geotextile and 300 mm thickness of topsoil on the top. According to [48], the soil below wetlands must be sufficiently impermeable in order to maintain wet conditions. The function of geotextiles was to avoid any soil particles from being washed out together with wastewater, which could cause clogging at the outlet pipe. All of the treatments were conducted in triplicate with 14 selected species of tropical plants, and one (1) set in triplicate was prepared without any plants as the control.

Domestic sewage consisting of black and grey water discharged from the student hostels of the Universiti Sains Malaysia (USM) Engineering Campus, Nibong Tebal, Penang, Malaysia, was used in this study. Domestic wastewater was pumped from the manhole and stored in an inlet tank. Domestic wastewater characterisation was performed before the mesocosm experiments to determine the nutrient and heavy metal content.

Homogenous domestic sewage stored in the inlet tank was first gravity flowed into a 45 L container for accurate volume measurement and subsequently dosed into the wetland mesocosm. The initial depth of the wastewater in the mesocosm system was set at 300 mm from the topsoil's surface. The initial water depth was limited by the height of the column and the available space above the soil surface. The wastewater was released until a depth of 150 mm for water sample collection, so as to ensure that the soil in the column would still be inundated with wastewater and to allow room for the next wastewater dosing. The effluent was collected from the mesocosm outlet after 48 h of retention. For the heavy metal analysis, soil and plant samples from the leaves, stems, and roots were also collected. The experimental setup is shown in Figure 1, while Figure 2 shows the experimental mesocosm layout.

### 2.2. Tropical Wetland Plant Selection and Cultivation

Fourteen (14) species of common and easily obtained wetland plants were selected in the study, as listed in Table 1. The plants were cultivated in polybags for three (3) weeks and in basins for floating plants before transplanting them into the mesocosms. The plants were then trimmed to 400 mm above the media surface and allowed to acclimatise for two (2) weeks in the columns before the experiments started. Some of the plant species were selected based on their rapid growth characteristics in order to aid in fast nutrient uptake from domestic sewage, which could be invasive if not native to the area.

### 2.3. Sample Collection and Preservation

Plant samples of the leaves, stems, and roots were harvested, cleaned, and oven-dried. Soil samples from the mesocosms were collected and stored in clean vinyl bags to prevent contamination. The soil samples were oven-dried at 50 °C and passed through a 2 mm sieve. The dried soil samples were further powdered and homogenised with a mortar and pestle for further analysis.

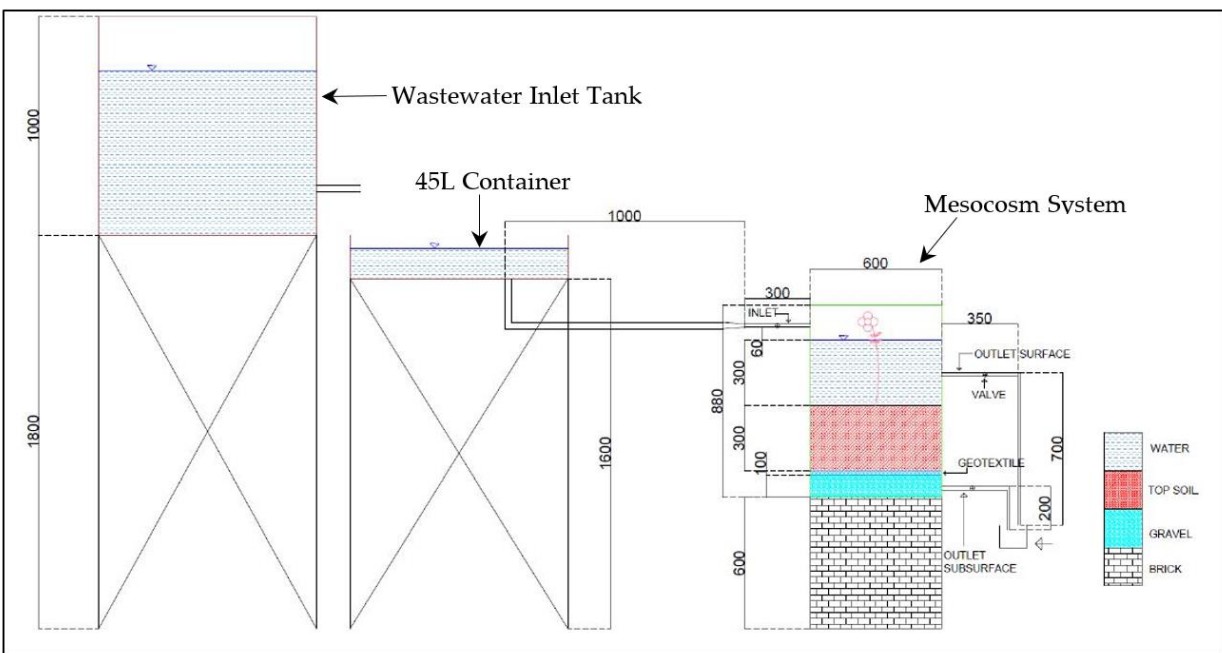

**Figure 1.** Wetland mesocosm experimental setup. The experimental setup consists of a wastewater inlet tank, 45 L containers, and 45 units of mesocosm columns.

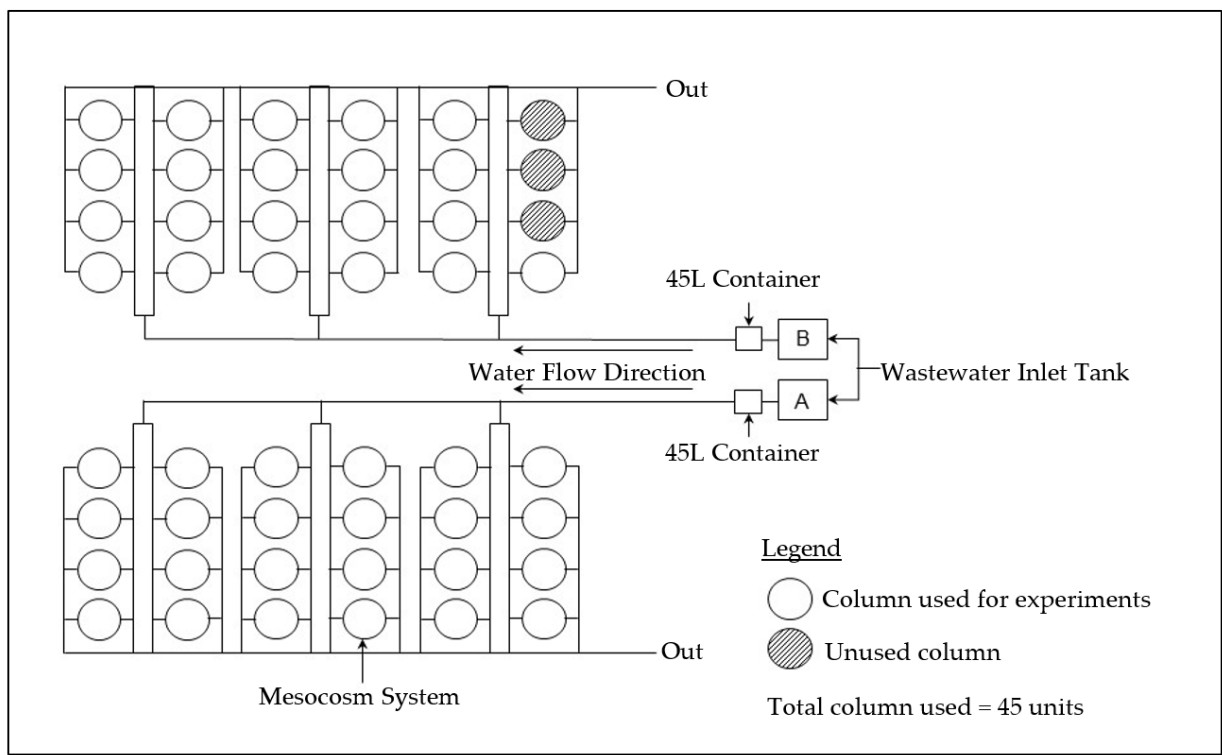

**Figure 2.** Mesocosm system layout. Because of the limited space for the experimental setup, domestic sewage will flow under gravity from separate wastewater inlets of Tank A and Tank B to two different sets of columns. The total columns used in the experiments are 45 units, with three columns used as the control without plants. The diameter for each column is 600 mm.

**Table 1.** Plant species selected for the wetland mesocosm study.

| Code No. | Common Name | Scientific Name | Plant Type |
| --- | --- | --- | --- |
| W1 | Tall Sedge | *Carex appressa* | Sedge |
| W2 | Vetiver Grass | *Chrysopogon zizanioides* | Perennial Grass |
| W3 | Common Spikerush | *Eleocharis dulcis* | Perennial Herb |
| W4 | Cattail | *Typha angustifolia* | Perennial |
| W5 | False Bird of Paradise | *Heliconia psittacorum* | Perennial |
| W6 | Blue Water Hyssop | *Bacopa caroliniana* | Perennial (Emergent/Submerged) |
| W7 | Alligator-flag | *Thalia geniculata* | Perennial |
| W8 | Canna Lily | *Canna x generalis* | Perennial |
| W9 | Water Mimosa * | *Neptunia oleracea* | Floating Leaves |
| W10 | Yam | *Colocasia esculenta* | Perennial |
| W11 | Mexican Sword | *Echinodorus palifolius* | Perennial |
| W12 | Water Hyacinth * | *Eichhornia crassipes* | Free-floating |
| W13 | Water Spinach * | *Ipomoea aquatica* | Floating Leaves |
| W14 | Giant Salvinia * | *Salvinia molesta* | Free-floating |

Note: * invasive species [48].

### 2.4. Plant and Soil Digestion

The samples were digested following the aqua regia digestion method for heavy metal concentration analysis using a CEM MDS-2000 microwave oven. This study used the digestion method suggested by [49]. The aqua regia solution was prepared by mixing 130 mL of concentrated HCl with 120 mL of water and then adding 150 mL of this solution to 50 mL of concentrated $HNO_3$ [50]. The samples were subsequently immersed in 15 mL of aqua regia solution overnight. For the aqua regia digestion, the samples to be analysed were randomly drawn from a well-mixed sample of 1 g and weighed into a 120-mL Teflon-PFA microwave digestion vessel. The samples were digested using the microwave digester by heating the samples at 200 °C for 40 min. After digestion, the samples were cooled at room temperature. The samples were then filtered through Whatman no. 541 filters, transferred to 50 mL volumetric flasks, and topped up with 0.25 M $HNO_3$ to the mark. Then, the samples were analysed for metals (Ca, Fe, Mg, and Mn) using Inductively Coupled Plasma Optical Emission Spectroscopy (ICP-OES).

### 2.5. Bioconcentration Factor (BCF) and Translocation Factor (TF) for Plant Data Analysis

Data on the heavy metal concentrations in plants were used to calculate the bioconcentration and translocation factors. The tendency of plants to accumulate metals from the substrate can be determined using the bioconcentration factor (BCF). The BCF values can be observed in sediments and plant organs such as roots, stems, and leaves. The bioconcentration factor (BCF) can be calculated based on the following equation [51]:

$$\text{Bioconcentration factor (BCF)} = \frac{\text{Concentration of metal in plant tissue} \left(\text{mgkg}^{-1}\right)}{\text{Concentration of metal in soil} \left(\text{mgkg}^{-1}\right)} \quad (1)$$

The translocation factor (TF) is used to determine the potential of plants for phytoremediation purposes. It can be calculated from the ratio of the examined metal concentration in the leaves or stems compared with the examined metal in the plant roots [46] based on the following equation:

$$\text{Translocation factor (TF)} = \frac{\text{Concentration of metal in leaves or stems} \left(\text{mgkg}^{-1}\right)}{\text{Concentration of metal in roots} \left(\text{mgkg}^{-1}\right)} \quad (2)$$

Plants with both a phytostabilisation and metal-tolerance capacity could be useful for phytoremediation. Both BCF and TF are essential for assessing the feasibility of a plant

species for phytoremediation purposes [51]. A BCF value of more than 1 demonstrates the potential success of a plant species for phytoremediation [52,53]. Plants with both BCF and TF values greater than 1 can be used as phytoextractors [54], whereas plants with a BCF value of greater than 1 and a TF value lower than 1 are phytostabilisers [55].

*2.6. Enrichment Factor (EF) and Geoaccumulation Index (Igeo) for Soil Data Analysis*

The enrichment factor (EF) and the geo-accumulation index ($I_{geo}$) are commonly utilised to analyse the metal concentrations in the soil. This universal index is a simple and quick method for assessing the extent of the enrichment and for analysing pollution levels across different environmental sources. Metal assessment and the degree of pollution were determined by calculating EF and $I_{geo}$ [56]. The enrichment factor (EF) can be calculated based on the equation below:

$$\text{Enrichment Factor (EF)} = \frac{[\text{Cn/Cref}]}{[\text{Bn/Bref}]} \tag{3}$$

where

$C_n$ is the concentration of the measured metal in the studied soil (mgkg$^{-1}$);

$C_{ref}$ is the concentration of the measured metal in the background environment (mgkg$^{-1}$);

$B_n$ is the concentration of the reference metal in the studied soil (mgkg$^{-1}$);

$B_{ref}$ is the concentration of the reference metal in the background environment (mgkg$^{-1}$).

EF can be categorised into six classifications, as shown in Table 2 [50,57,58].

**Table 2.** Classification of enrichment factor (EF).

| Enrichment Factor (EF) | Degree of Enrichment |
|---|---|
| EF < 2 | Depletion to minimal enrichment (no or minimal pollution) |
| 2 ≤ EF < 5 | Moderate enrichment (moderate pollution) |
| 5 ≤ EF < 20 | Significant enrichment (significant pollution) |
| 20 ≤ EF < 40 | Very high enrichment (very strong pollution) |
| EF > 40 | Extreme enrichment (extreme pollution) |

The geoaccumulation index ($I_{geo}$) determines the degree of metal accumulation in soils and can be obtained using the following equation:

$$\text{Geoaccumulation Index, Igeo} = \log_2\left[\frac{\text{Cn}}{1.5\text{Bn}}\right] \tag{4}$$

where:

$C_n$ is the concentration of measured metal in the studied soil (mgkg$^{-1}$);

$B_n$ is the reference value of the measured metal (mgkg$^{-1}$).

The index was initially defined by [59], and the soil quality was classified into several classes, as seen in Table 3.

**Table 3.** Classification of the geoaccumulation index.

| Igeo | Igeo Class | Description of Soil Quality |
|---|---|---|
| <0 | 0 | Uncontaminated |
| 0–1 | 1 | Uncontaminated to moderately contaminated |
| 1–2 | 2 | Moderately contaminated |
| 2–3 | 3 | Moderately to strongly contaminated |
| 3–4 | 4 | Strongly contaminated |
| 4–5 | 5 | Strongly to extremely strongly contaminated |
| >5 | 6 | Extremely contaminated |

In this study, iron (Fe) was chosen as a reference element or metal due to its association with a fine solid surface, being uniformly distributed in natural soil, and its geochemistry being close to many trace elements [50,60]. Other reference elements used include aluminium (Al) [57,61], manganese (Mn), scandium (Sc), and zinc (Zn) [62,63]. The background metal concentrations (Ca, Fe, Mg, and Mn) used in this study were those reported by [64] for the upper continental crust, as there was no detailed information available on the background metal concentrations for soil in Peninsular Malaysia, even though the authors of [65] reported some of the background levels of heavy metal concentration in sediments sampled along the west coast of Peninsular Malaysia.

### 2.7. Statistical Analysis

Statistical analysis of one-way ANOVA with post hoc multiple comparisons test using Tukey's method was performed with IBM SPSS Statistics Version 25 software (IBM, New York, NY, USA).

### 3. Results and Discussion

#### 3.1. Bioconcentration Factor (BCF)

In this mesocosm study, a total of 14 different tropical plant species were applied in the phytoremediation of selected metals in domestic sewage. The BCF values for various tropical plants of selected metals consisting of calcium (Ca), iron (Fe), magnesium (Mg), and manganese (Mn) will be discussed in this sub-section.

Figure 3 shows the BCF values calculated from the phytoremediation of Ca by 14 different tropical plant species. It was observed that all plant species depicted BCF values higher than 1 for Ca, indicating that all of the plants studied had the potential for the phytoremediation of Ca. A larger value of BCF implies a better phytoaccumulation capability [66]. W12 resulted in the highest $BCF_{mean}$ value of 21.10 $\pm$ 5.04, followed by W6 ($BCF_{mean}$ 17.14 $\pm$ 3.33), W4 ($BCF_{mean}$ 12.40 $\pm$ 4.65), W5 ($BCF_{mean}$ 11.69 $\pm$ 2.12), and W11 ($BCF_{mean}$ 10.47 $\pm$ 6.63). We found that plants such as W6 ($BCF_{leave}$ 20.50), W11 ($BCF_{leave}$ 18.07), and W4 ($BCF_{leave}$ 17.15) better accumulated Ca in their leaves, whereas plants such as W12 and W5 accumulated Ca in their stems ($BCF_{stem}$ 26.46 and $BCF_{stem}$ 13.87). W9 demonstrated better accumulation of Ca in the roots ($BCF_{root}$ 16.57). One-way ANOVA revealed that the $BCF_{mean}$ was significantly different at $p = 0.000$ between all of the plant species tested. However, Tukey's HSD multiple comparison tests of the mean BCF showed that W12, with the highest $BCF_{mean}$, was not significantly different from W6, W4, W5, and W11, indicating that all the top five plant species possessed the same capability for Ca bioaccumulation.

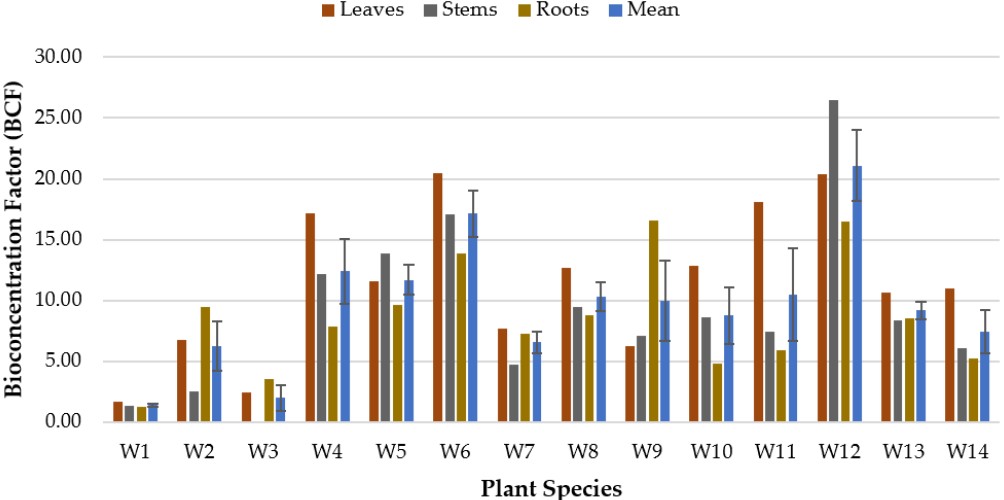

**Figure 3.** Bioconcentration factor (BCF) of calcium (Ca) in the phytoremediation of domestic sewage mesocosms using various tropical plants.

As for the phytoremediation of Fe (Figure 4), all of the plant species depicted $BCF_{mean}$ values lower than 1. However, W13, W6, and W11 had higher $BCF_{root}$ values (>1) in their roots ($BCF_{root}$ 2.65, 1.25, and 1.23, respectively). This shows that only the roots of these plants showed the potential to uptake Fe from the soil added with domestic sewage. One-way ANOVA showed no significant difference ($p = 0.876$) between the $BCF_{mean}$ of various plant species, indicating that there were no plant species with significant bioaccumulation properties even at low values.

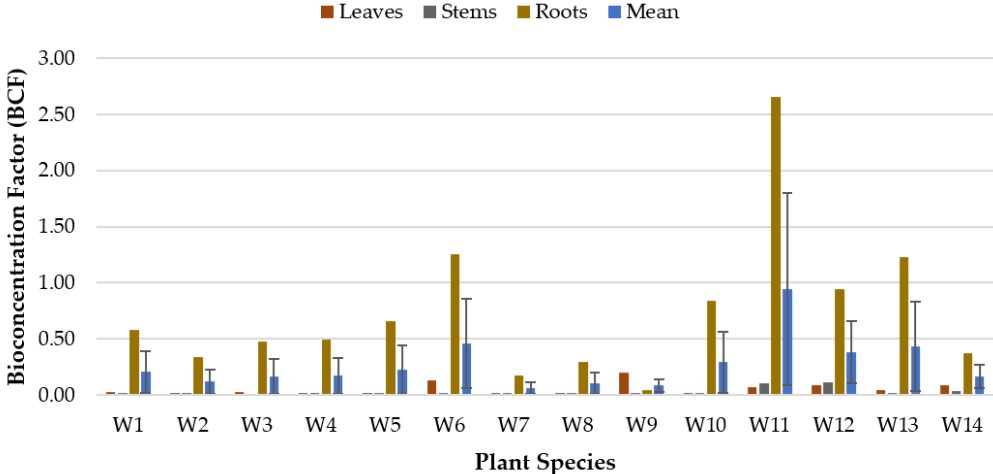

**Figure 4.** Bioconcentration Factor (BCF) of Iron (Fe) in phytoremediation of domestic sewage mesocosms using various tropical plants.

As for the phytoremediation study of Mg (Figure 5), all of the plant species demonstrated BCF values higher than 1 for all of the plant parts (except the roots for W6 and stems for W7), indicating that almost all plant species were capable of accumulating Mg in almost every part of their plant tissues (leaves, stems, and roots). W11 exhibited the highest potential to accumulate Mg in the roots with a $BCF_{root}$ value of 227.44, followed by W13 ($BCF_{root}$ 105.56), W10 ($BCF_{root}$ 96.10), W12 ($BCF_{root}$ 80.89), W8 ($BCF_{root}$ 35.39), and W14 ($BCF_{root}$ 31.61). W7 accumulated Mg better in the leaves than other plant species ($BCF_{leave}$ 30.74), whereas W5 demonstrated a uniform or balanced accumulation of Mg in all plant parts (leaves, stems, and roots) with $BCF_{leave}$, $BCF_{stem}$, and $BCF_{root}$ values of 30.43, 32.91, and 32.55, respectively. One-way ANOVA tested using $BCF_{mean}$ values displayed that there was no significant difference ($p = 0.708$) between all of the plant species, suggesting that all plants were capable of bioaccumulating Mg at various concentrations in all of the plant parts.

The $BCF_{leave}$, $BCF_{stem}$, $BCF_{root}$, and $BCF_{mean}$ values of Mn from various plants in the domestic sewage-treated soil are depicted in Figure 6. It was observed that 10 plant species recorded BCF values higher than 1 in at least one part of the plants after the addition of domestic sewage. Four plant species, namely W1, W2, W4, and W5, displayed BCF values higher than 1 in all of their leaves, stems, and roots. Among these plants, W5 depicted the highest $BCF_{mean}$ value of 21.70 ± 13.38 for Mn, followed by W4 ($BCF_{mean}$ 6.04 ± 4.94), W1 ($BCF_{mean}$ 3.08 ± 2.02), and W2 ($BCF_{mean}$ 2.59 ± 1.53). One-way ANOVA showed that the $BCF_{mean}$ values were significantly different between all of the plant species tested ($p < 0.05$). Tukey's HSD multiple comparison tests were analysed on the four plant species with BCF values higher than 1 in all of the plant parts. It was revealed that W5, with the highest $BCF_{mean}$, was significantly different from W4 ($p = 0.05$), W1 ($p = 0.01$), and W2 ($p = 0.007$), indicating that W5 was a better Mn bioaccumulator compared with the other plant species.

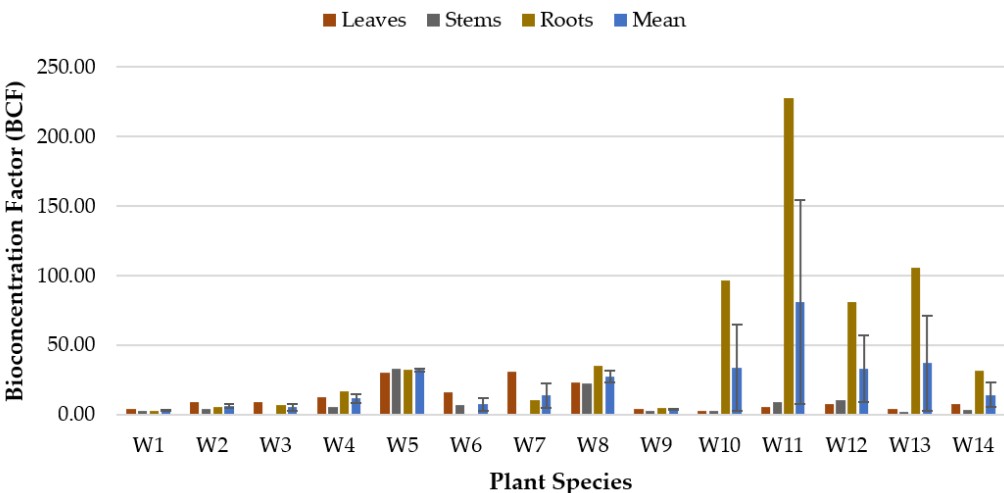

**Figure 5.** Bioconcentration factor (BCF) of magnesium (Mg) in the phytoremediation of domestic sewage mesocosms using various tropical plants.

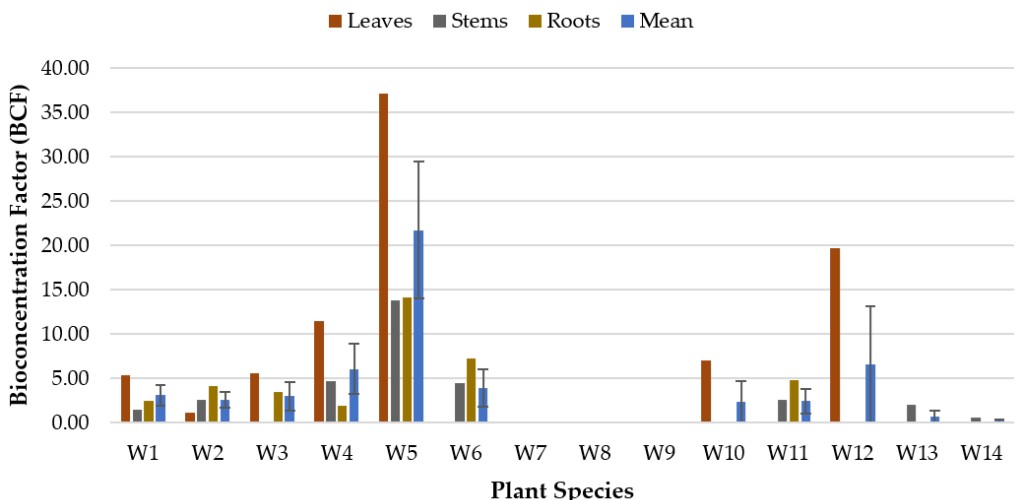

**Figure 6.** Bioconcentration Factor (BCF) of Manganese (Mn) in phytoremediation of domestic sewage mesocosms using various tropical plants.

### 3.2. Translocation Factor (TF)

The $TF_{leave}$, $TF_{stem}$, and $TF_{mean}$ values for various tropical plants considering the examined metals, consisting of calcium (Ca), iron (Fe), magnesium (Mg), and manganese (Mn), from domestic sewage phytoremediation mesocosms are presented in this sub-section. TF was used to determine a plant's potential for the translocation of metals from the roots to the leaves [67]. A larger value for TF implies a higher translocation capability [66]. A TF value greater than 1 is the metal accumulator, whereas a value less than 1 is the metal excluder species [51,68]. It was observed that almost all plants recorded a TF value of Ca greater than 1 in both the leaves and stems, except a few plants, such as W2, W3, and W9, with values lower than 1 (Figure 7). W11 depicted the highest $TF_{leave}$ value of 3.07 in the leaves, followed by W10 ($TF_{leave}$ 2.69), W4 ($TF_{leave}$ 2.18), and W14 ($TF_{leave}$ 2.08) after the addition of domestic wastewater into the mesocosms, whereas W10 recorded the highest $TF_{stem}$ value of 1.81 in the stems, followed by W12 ($TF_{stem}$ 1.61), W4 ($TF_{stem}$ 1.55), and W5 ($TF_{stem}$ 1.44).

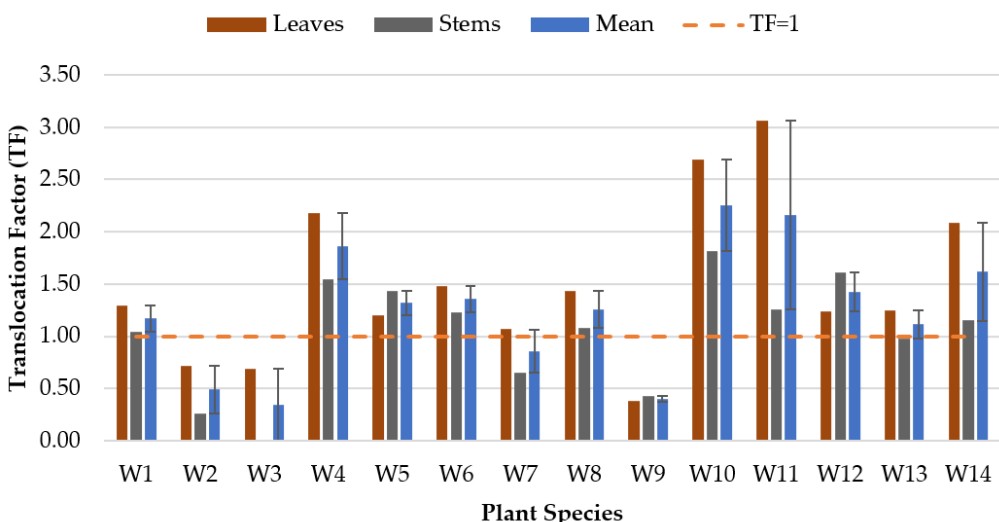

**Figure 7.** Translocation factor (TF) of calcium (Ca) in the phytoremediation of domestic sewage mesocosms using various tropical plants.

Figure 8 shows the $TF_{leave}$, $TF_{stem}$, and $TF_{mean}$ values of Fe in various tropical plants treated with domestic wastewater in the mesocosms. It was observed that the transportation of Fe within the plants was very weak, as almost all of the TF values were below one, except for W9 leaves with a $TF_{leave}$ value of 4.80. As for the translocation of Mg in the tropical plants (Figure 9), only leaves from W1 ($TF_{leave}$ 1.34), W2 ($TF_{leave}$ 1.82), W3 ($TF_{leave}$ 1.33), and W7 ($TF_{leave}$ 2.94), as well as W5 stems ($TF_{stem}$ 1.01), depicted values greater than 1, thus indicating that the translocation of Mg to the parts of these plants was strong. According to Figure 10 for the translocation of Mn, it was observed that only plants species of W1, W3, W4, and W5 recorded TF values of more than 1 in either its leaves, stems or both, with W4 depicting the highest $TF_{mean}$ value of 4.26 ± 1.79.

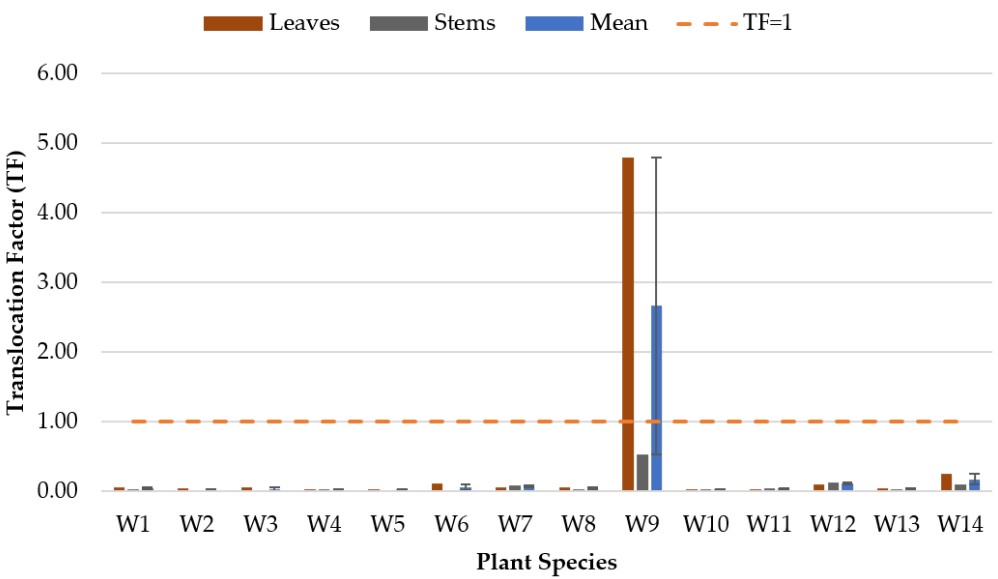

**Figure 8.** Translocation factor (TF) of iron (Fe) in the phytoremediation of domestic sewage mesocosms using various tropical plants.

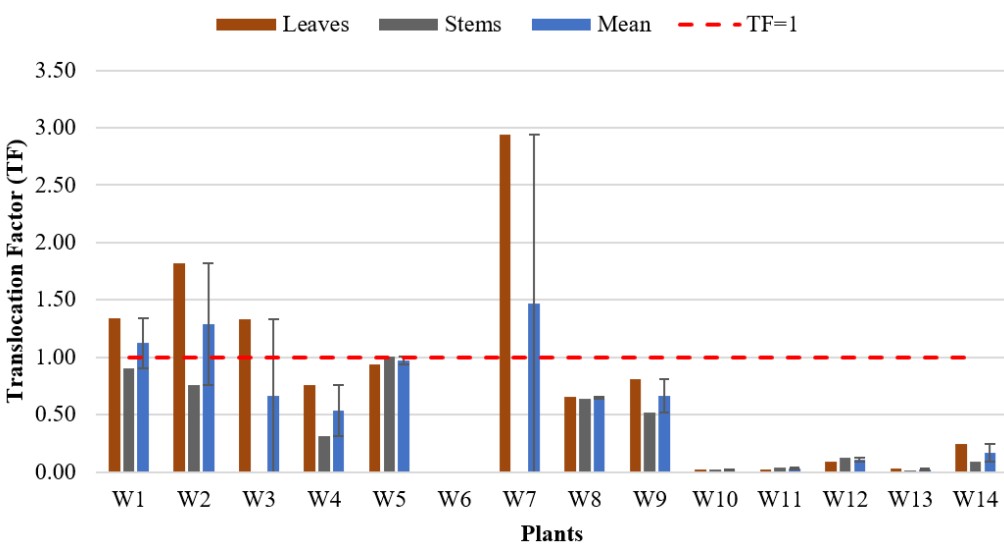

**Figure 9.** Translocation factor (TF) of magnesium (Mg) in the phytoremediation of domestic sewage mesocosms using various tropical plants.

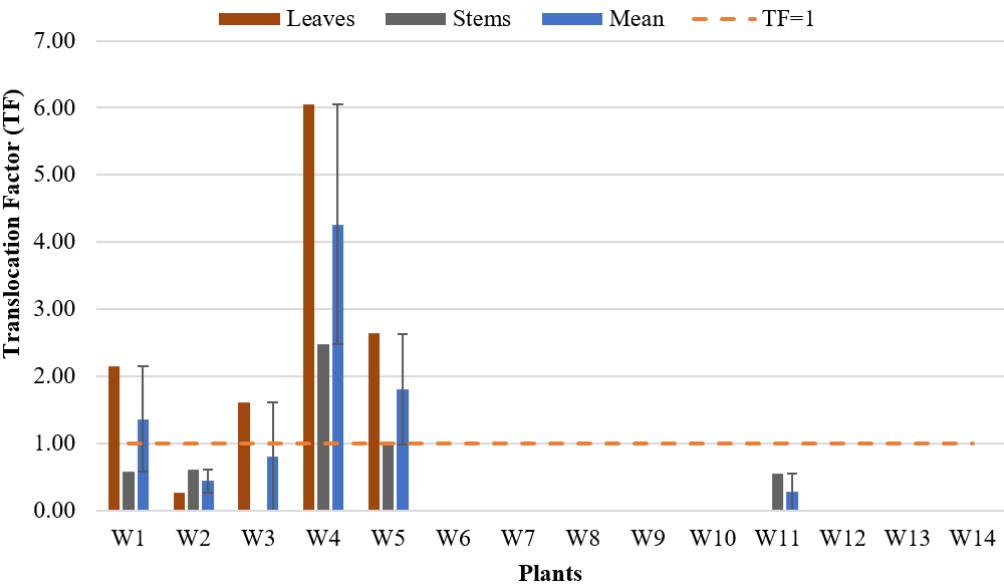

**Figure 10.** Translocation factor (TF) of manganese (Mn) in the phytoremediation of domestic sewage mesocosms using various tropical plants.

### 3.3. Phytoextration and Phytostabilisation Property

Plants for phytoremediation can be categorised as phytoextractors when both BCF and TF values are greater than 1 [49]. In contrast, plants with a BCF value of greater than 1 and a TF value of lower than 1 are phytostabilisers [50]. From the BCF and TF values discussed in Sections 3.1 and 3.2, the suggested top five Ca accumulator plant species, consisting of W12, W6, W4, W5, and W11, depicted Ca phytoextraction properties, as these plants had Ca $BCF_{mean}$ and $TF_{mean}$ values higher than 1. As for the accumulation of Fe, the suggested plant species as good Fe phytostabilisers were W13, W6, and W11, as the $TF_{mean}$ values were less than 1, whereas the $BCF_{root}$ values were more significant than 1, as the translocation of Fe from the roots to the leaves in these plants was low and at the same time capable of uptaking Fe from the soil. Fe is an essential element for plant growth [69]. Although Fe is the sixth most abundant element in the universe and the fourth most abundant in the Earth's crust [70], its availability to plant roots is very low as Fe

availability is dictated by soil conditions such as the soil redox potential and pH, where aerobic conditions and a higher pH will cause Fe to oxidise into insoluble ferric oxides [69]. Moreover, Fe availability in the soil can also be affected by soil compaction, soil moisture, temperature, and levels of phosphorus, nitrogen, zinc, manganese, and potassium in the soil [71]. A high accumulation of metals in the roots and the minimum translocation of metals to other plant tissues might be possible due to the sequestration of metals inside the root vacuoles of the plant, where metals are fixed as nontoxic elements [72]. For the phytoremediation of Mg, W7, W2, and W1 depicted phytoextraction properties as both $BCF_{mean}$ and $TF_{mean}$ were greater than 1. Other Mg phytoremediators such as W11, W13, W10, W12, W5, W8, W14, W6, W3, and W9 depicted phytostabilisation properties as the $BCF_{mean}$ values were greater than 1, whereas the $TF_{mean}$ values were lower than 1. The suggested plant species for Mn phytoextraction are W5, W4, and W1, as the $BCF_{mean}$ and $TF_{mean}$ values were greater than 1. In contrast, the potential Mn phytostabilisation plant species were W3, W2, and W11 as the $BCF_{mean}$ values were greater than 1, but the $TF_{mean}$ values were lower than 1. However, bioaccumulation and metal accumulation in plant species vary from metal to metal and species to species [73]. A pattern of bioaccumulation and translocation of metals from the roots to shoots needs to be established in tropical plants, which could be beneficial for treating domestic wastewater through phytoremediation and for selecting tolerant plant species. The plant species with various phytoremediation properties as well as their respective $BCF_{mean}$ and $TF_{mean}$ values are summarised in Table 4.

**Table 4.** Summary of plant species based on various types of phytoremediation properties.

| Metals | Phytoextraction | | | Phytostabilisation | | |
|---|---|---|---|---|---|---|
| | $BCF_{mean}$ | $TF_{mean}$ | Plant Species | $BCF_{mean}$ | $TF_{mean}$ | Plant Species |
| Calcium (Ca) | $21.10 \pm 5.04$ | $1.42 \pm 0.18$ | Water Hyacinth (*Eichhornia crassipes*) | $9.97 \pm 5.72$ | $0.40 \pm 0.02$ | Water Mimosa (*Neptunia oleracea*) |
| | $17.14 \pm 3.33$ | $1.36 \pm 0.12$ | Blue Water Hyssop (*Bacopa caroliniana*) | $6.54 \pm 1.61$ | $0.86 \pm 0.21$ | Alligator-flag (*Thalia geniculata*) |
| | $12.40 \pm 4.65$ | $1.87 \pm 0.32$ | Cattail (*Typha angustifolia*) | $6.26 \pm 3.52$ | $0.49 \pm 0.22$ | Vetiver Grass (*Chrysopogon zizanioides*) |
| | $11.69 \pm 2.12$ | $1.32 \pm 0.12$ | False Bird of Paradise (*Heliconia psittacorum*) | $1.99 \pm 1.81$ | $0.34 \pm 0.34$ | Common Spikerush (*Eleocharis dulcis*) |
| | $10.47 \pm 6.63$ | $2.16 \pm 0.90$ | Mexican Sword (*Echinodorus palifolius*) | - | - | - |
| | $10.32 \pm 2.05$ | $1.26 \pm 0.18$ | Canna Lily (*Canna x generalis*) | - | - | - |
| | $9.18 \pm 1.29$ | $1.11 \pm 0.14$ | Water Spinach (*Ipomoea aquatica*) | - | - | - |
| | $8.75 \pm 4.03$ | $2.25 \pm 0.44$ | Yam (*Colocasia esculenta*) | - | - | - |
| | $7.42 \pm 3.08$ | $1.62 \pm 0.47$ | Giant Salvinia (*Salvinia molesta*) | - | - | – |
| | $1.41 \pm 0.20$ | $1.17 \pm 0.12$ | Tall Sedge (*Carex appressa*) | - | - | |
| Iron (Fe) | - | - | - | 2.652 * | $0.03 \pm 0.01$ | Mexican Sword (*Echinodorus palifolius*) |
| | - | - | - | 1.252 * | $0.05 \pm 0.05$ | Blue Water Hyssop (*Bacopa caroliniana*) |
| | - | - | - | 1.231 * | $0.03 \pm 0.01$ | Water Spinach (*Ipomoea aquatica*) |

**Table 4.** *Cont.*

| Metals | Phytoextraction | | | Phytostabilisation | | |
|---|---|---|---|---|---|---|
| | BCF$_{mean}$ | TF$_{mean}$ | Plant Species | BCF$_{mean}$ | TF$_{mean}$ | Plant Species |
| Magnesium (Mg) | 13.73 ± 15.63 | 1.47 ± 1.47 | Alligator-flag (*Thalia geniculata*) | 80.80 ± 127.01 | 0.03 ± 0.01 | Mexican Sword (*Echinodorus palifolius*) |
| | 6.07 ± 2.83 | 1.29 ± 0.53 | Vetiver Grass (*Chrysopogon zizanioides*) | 37.04 ± 59.35 | 0.03 ± 0.01 | Water Spinach (*Ipomoea aquatica*) |
| | 3.06 ± 0.65 | 1.12 ± 0.22 | Tall Sedge (*Carex appressa*) | 33.68 ± 54.06 | 0.03 ± 0.00 | Yam (*Colocasia esculenta*) |
| | - | - | - | 32.81 ± 41.66 | 0.11 ± 0.22 | Water Hyacinth (*Eichhornia crassipes*) |
| | - | - | - | 31.96 ± 1.34 | 0.97 ± 0.04 | False Bird of Paradise (*Heliconia psittacorum*) |
| | - | - | - | 27.13 ± 7.16 | 0.65 ± 0.01 | Canna Lily (*Canna x generalis*) |
| | - | | - | 14.08 ± 15.36 | 0.17 ± 0.07 | Giant Salvinia (*Salvinia molesta*) |
| | - | - | - | 11.53 ± 5.83 | 0.54 ± 0.22 | Cattail (*Typha angustifolia*) |
| | - | - | - | 7.40 ± 7.89 | 0.00 ± 0.00 | Blue Water Hyssop (*Bacopa caroliniana*) |
| | - | - | - | 5.23 ± 4.66 | 0.67 ± 0.67 | Common Spikerush (*Eleocharis dulcis*) |
| | - | - | - | 3.67 ± 1.14 | 0.67 ± 0.14 | Water Mimosa (*Neptunia oleracea*) |
| Manganese (Mn) | 21.70 ± 13.38 | 1.81 ± 0.83 | False Bird of Paradise (*Heliconia psittacorum*) | 3.00 ± 2.81 | 0.81 ± 0.81 | Common Spikerush (*Eleocharis dulcis*) |
| | 6.04 ± 4.94 | 4.26 ± 1.79 | Cattail (*Typha angustifolia*) | 2.59 ± 1.53 | 0.44 ± 0.18 | Vetiver grass (*Chrysopogon zizanioides*) |
| | 3.08 ± 2.02 | 1.36 ± 0.79 | Tall Sedge (*Carex appressa*) | 2.44 ± 2.38 | 0.27 ± 0.27 | Mexican Sword (*Echinodorus palifolius*) |

Notes: * All BCF values shown in the table are mean values, except for the metal iron (Fe), which shows the BCF values from the roots.

*3.4. Enrichment Factor (EF) and Geoaccumulation Index ($I_{geo}$)*

Table 5 shows the results of the enrichment factor (EF) values for soil before and after the phytoremediation of domestic wastewater with 14 tropical plant species. As EF was calculated using Fe as a reference metal, the EF value for Fe was 1.0. It was observed that the EF for all of the metals (Ca, Mg, and Mn) in all of the soil samples decreased after the phytoremediation of the domestic wastewater experiment. The EF values for all of the metals were less than 2, which indicated depletion to mineral enrichment according to the EF classification in Table 2.

Similarly, the geoaccumulation index ($I_{geo}$) also depicted a decrease in metals accumulation in the soil after the experiment (Table 6). In general, all of the soil samples were classified as uncontaminated based on the $I_{geo}$ indices in Table 3. Based on these two indices, it can be suggested that the plants may have facilitated in the removal of these metals from the soil through uptake into the plant tissues from the roots.

**Table 5.** Enrichment Factor (EF) for soil before and after phytoremediation of domestic sewage with various tropical plant species.

| Plant Species | Enrichment Factor (EF) | | | | | | | |
|---|---|---|---|---|---|---|---|---|
| | Calcium (Ca) | | Iron (Fe) | | Magnesium (Mg) | | Manganese (Mn) | |
| | Before | After | Before | After | Before | After | Before | After |
| W1 | 0.086 | 0.026 | 1.000 | 1.000 | 0.070 | 0.006 | 0.354 | 0.038 |
| W2 | 0.086 | 0.008 | 1.000 | 1.000 | 0.070 | 0.004 | 0.354 | 0.023 |
| W3 | 0.086 | 0.026 | 1.000 | 1.000 | 0.070 | 0.006 | 0.354 | 0.038 |
| W4 | 0.086 | 0.026 | 1.000 | 1.000 | 0.070 | 0.006 | 0.354 | 0.038 |
| W5 | 0.086 | 0.026 | 1.000 | 1.000 | 0.070 | 0.006 | 0.354 | 0.038 |
| W6 | 0.086 | 0.026 | 1.000 | 1.000 | 0.070 | 0.006 | 0.354 | 0.038 |
| W7 | 0.086 | 0.026 | 1.000 | 1.000 | 0.070 | 0.006 | 0.354 | 0.038 |
| W8 | 0.086 | 0.026 | 1.000 | 1.000 | 0.070 | 0.006 | 0.354 | 0.038 |
| W9 | 0.086 | 0.074 | 1.000 | 1.000 | 0.070 | 0.018 | 0.354 | 0.053 |
| W10 | 0.086 | 0.074 | 1.000 | 1.000 | 0.070 | 0.018 | 0.354 | 0.053 |
| W11 | 0.086 | 0.063 | 1.000 | 1.000 | 0.070 | 0.024 | 0.354 | 0.095 |
| W12 | 0.086 | 0.063 | 1.000 | 1.000 | 0.070 | 0.024 | 0.354 | 0.095 |
| W13 | 0.086 | 0.063 | 1.000 | 1.000 | 0.070 | 0.024 | 0.354 | 0.095 |
| W14 | 0.086 | 0.063 | 1.000 | 1.000 | 0.070 | 0.024 | 0.354 | 0.095 |

**Table 6.** Geoaccumulation index (Igeo) for soil before and after the phytoremediation of domestic sewage with various tropical plant species.

| Plant Species | Geoaccumulation Index ($I_{geo}$) | | | | | | | |
|---|---|---|---|---|---|---|---|---|
| | Calcium (Ca) | | Iron (Fe) | | Magnesium (Mg) | | Manganese (Mn) | |
| | Before | After | Before | After | Before | After | Before | After |
| W1 | −11.25 | −13.35 | −7.72 | −8.08 | −11.56 | −15.54 | −9.22 | −12.80 |
| W2 | −11.25 | −14.41 | −7.72 | −7.49 | −11.56 | −15.64 | −9.22 | −12.95 |
| W3 | −11.25 | −13.35 | −7.72 | −8.08 | −11.56 | −15.54 | −9.22 | −12.80 |
| W4 | −11.25 | −13.35 | −7.72 | −8.08 | −11.56 | −15.54 | −9.22 | −12.80 |
| W5 | −11.25 | −13.35 | −7.72 | −8.08 | −11.56 | −15.54 | −9.22 | −12.80 |
| W6 | −11.25 | −13.35 | −7.72 | −8.08 | −11.56 | −15.54 | −9.22 | −12.80 |
| W7 | −11.25 | −13.35 | −7.72 | −8.08 | −11.56 | −15.54 | −9.22 | −12.80 |
| W8 | −11.25 | −13.35 | −7.72 | −8.08 | −11.56 | −15.54 | −9.22 | −12.80 |
| W9 | −11.25 | −12.34 | −7.72 | −8.57 | −11.56 | −14.39 | −9.22 | −12.80 |
| W10 | −11.25 | −12.34 | −7.72 | −8.57 | −11.56 | −14.39 | −9.22 | −12.80 |
| W11 | −11.25 | −13.10 | −7.72 | −9.10 | −11.56 | −14.50 | −9.22 | −12.49 |
| W12 | −11.25 | −13.10 | −7.72 | −9.10 | −11.56 | −14.50 | −9.22 | −12.49 |
| W13 | −11.25 | −13.10 | −7.72 | −9.10 | −11.56 | −14.50 | −9.22 | −12.49 |
| W14 | −11.25 | −13.10 | −7.72 | −9.10 | −11.56 | −14.50 | −9.22 | −12.49 |

## 4. Conclusions

The potential of 14 tropical wetland plant species to remove heavy metals (Ca, Mg Fe, and Mn) from domestic sewage was examined through the bioconcentration factor (BCF), translocation factor (TF), enrichment factor (EF), and geoaccumulation factor (Igeo). The results indicate that 11 out of 14 species were magnesium phytostabilisers, 10 were calcium phytoextractors, and no plant species demonstrated ferrum phytoextraction properties. In general, three species were found to be good phytostabilisers, namely Water Mimosa, Alligator-flag, Vetiver Grass, and Common Spikerush, as they had a BCF > 1 and TF < 1.

For Fe phytoremediation, three species depicted Fe phytostabilisation properties: Mexican Sword, Blue Water Hyssop, and Water Spinach. The suggested plants for Mg phytoextraction (BCF > 1 and TF > 1) are Alligator-flag, Vetiver Grass, and Tall Sedge. As for Mn phytoremediation, three (3) plants displayed phytoextraction properties in the order of False Bird of Paradise > Cattail > Tall Sedge, whereas Common Spikerush, Vetiver grass, and Mexican Sword were found to have phytostabilisation properties.

The enrichment factor for all of the metals (Ca, Mg, and Mn) with Fe as a reference metal in all of the soil samples decreased after the phytoremediation of the domestic wastewater experiment, indicating depletion to mineral enrichment (EF < 2). Overall, all of the soil samples were classified as uncontaminated based on the $I_{geo}$ indices. Based on the factors and indices obtained, it is suggested that the plants facilitated in removing these metals from the soil through uptake into the plant tissues from the roots.

**Author Contributions:** Conceptualisation, S.Y.A., H.W.G., H.H. and N.A.A.; writing—original draft preparation, S.Y.A.; writing—review and editing, H.W.G., N.A.A., H.H. and B.M.F.; project administration, H.W.G.; funding acquisition, N.A.Z. and Z.J. All authors have read and agreed to the published version of the manuscript.

**Funding:** This research was funded by the Government of Malaysia through the National Water Research Institute of Malaysia (NAHRIM) under the 11th Malaysia Plan Development Project P23 17000 001 0003 and the Ministry of Higher Education, Malaysia, under the Higher Institution Centre of Excellence (HICoE) research grant (311.PREDAC.4403901).

**Data Availability Statement:** Not applicable.

**Acknowledgments:** The authors would like to express their appreciation to the Government of Malaysia through the National Water Research Institute of Malaysia (NAHRIM) for funding this study under the 11th Malaysia Plan Development Project P23 17000 001 0003. This research was supported in part by the Ministry of Higher Education, Malaysia, under the Higher Institution Centre of Excellence (HICoE) research grant (311.PREDAC.4403901). Special thanks to the River Engineering and Urban Drainage Research Centre (REDAC), Universiti Sains Malaysia (USM), for providing the study site and laboratory facilities, as well as the Water Quality and Environment Research Centre, National Water Research Institute of Malaysia (NAHRIM), for the support to carry out this research.

**Conflicts of Interest:** The authors declare no conflict of interest.

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
