# Peer review of "Heavy Metals Removal from Domestic Sewage in Batch Mesocosm Constructed Wetlands using Tropical Wetland Plants"

_water, doi:10.3390/w15040797_

Round 1

Reviewer 1 Report

In this manuscript, the authors investigated the removal potential of 14 tropical wetland plants for heavy metals in domestic wastewater by analyzing the bioconcentration factor (BCF), translocation factor (TF), enrichment factor (EF), and geoaccumulation index (Igeo), which is important for the removal of heavy metals from domestic wastewater and provides a theoretical reference for the field of wastewater treatment in the tropics. However, there are still many issues in this paper that need to be addressed carefully by the authors. The following are some specific comments for the authors to consider:

1. In the introduction section, please illustrate the novelty and innovation of this study.

2In Line 102, please elaborate on the basis for selecting these four heavy metals.

3The format of the references cited in the text needs to be further revised.

4The conclusion should be accurate.

5The language of the paper should be further improved.

Reviewer 2 Report

I found this manuscript interesting and based on a well-conducted study, with relevant analysis and a good mesocosm estimation to test several plant species on heavy metals pollution treatment of sewage, a topic of both scientific and social current interests. The key results are well discussed and exposed, only the Introduction section needs in my opinion some minor revisions.

Keywords are too articulate, please try to synthesize and avoid the use of words already reported in the Title.

Lines 37-38: not only, sometimes also carbon balancing could be interesting, but also in transitional basins and wetlands involved, enrich this sentence about, also using more references, such as:

https://doi.org/10.3390/w14010108

https://doi.org/10.1016/j.marpolbul.2019.06.054

Lines 48-61: this period should briefly also consider the importance of some heavy metals for living organisms and ecosystems equilibriums when traces:

https://doi.org/10.1007/BF02867382

https://doi.org/10.1007/978-94-007-4470-7_7

Lines 75-86: I suggest moving this period to the Discussion section, and rephrasing the reference style in introducing them, for example "as reported by Goh and colleagues..", or "Goh et al. studied the..".

Best regards

The Reviewer

Reviewer 3 Report

The article has been prepared carefully and contains all the required parts, including abundant use of citations from other literature. I agree with the methods used. The design of the experiment can be accepted.

I have no serious objections to the article.

Typos need to be corrected, e.g. on line 42 - "sullage" instead of "sludge".

Furthermore, I have some questions or suggestions that could help the readers understand the intention of the authors, the chosen design parameters of the experiment and the chosen types of vegetation. And at the same time increase the application potential of the obtained valuable results.

Questions and suggestions about the article:

1.

On line 53 in the introductory research section, Cd, Cr, Hg, Ni, Pb and Zn plus another risky element – As – are correctly included among problematic heavy metals.

But in chapter 3. Results and discussion, the results of the analyses for the elements Ca, Fe, Mg, Mn, which are not essential risk elements but nutrients (Ca), are described in detail. Why were these elements selected for detailed description instead of other risky ones?

Even the title of the article contains "heavy metals", but e.g. Ca is not normally included in this group of elements.

Even though the stated results are in principle correct and interesting from the point of view of publication and further use in research (citations) and in practice, I recommend considering exchanging the results and graphs for any of the stated elements for the results and graphs for any of the main risky heavy metals (Cd, Hg, Cr, Pb).

If the experiment was focused on the elements listed in detail, is it possible to add, as I wrote above, why they were selected and not the selection of PVVs, which are also described as risky in the research section?

2.

The following parameters are given in the design of the experiment: 300 mm and 150 mm water level height. It would be appropriate to add why these level levels were chosen. Ideally, add references to the literature describing the design parameters chosen in this way.

3.

The soil was chosen as the substrate. This is an unusual substrate for artificial wetlands.

The article on lines 62 to 74 contains information describing artificial wetland technology, including the importance of substrate as a filter medium for the effective treatment of flowing wastewater. In practice, however, porous filter material, from sand to gravel, is used in flow-through wetlands. Soil as a substrate is more suitable for artificial wetlands with a free surface and surface flow, or for purified wastewater as the next stage of the technological line.

Therefore, it would be good to add the exact technology of the artificial wetland, the degree of cleaning, and the choice of soil as a substrate in the described experiment.

4.

Selection of free-floating plants for the experiment:

The selection of plants includes emergent plants, which are used in wetlands with water filtration through a porous medium, and free-floating species, which are practically used in wetlands with a free surface and surface water flow.

So does the described research direct the results to use in free surface wetlands with the surface flow?

5.

In the selection of plants, the question of their possible invasive spread is important, if they are not used in the areas where they are native. In the article, it would be appropriate to assess and supplement this fact for selected species. The information will be useful for potential users from regions other than Malaysia.

An example of an invasive species is the plant W12 – water hyacinth.
